# The Role of Sublobar Resection for the Surgical Treatment of Non-Small Cell Lung Cancer

Parnia Behinaein [1], John Treffalls [2], Hollis Hutchings [3] and Ikenna C. Okereke [3,*]

1    School of Medicine, Wayne State University, Detroit, MI 48202, USA; parnia.behinaein@wayne.edu
2    Long School of Medicine, University of Texas Health San Antonio, San Antonio, TX 78229, USA; treffalls@livemail.uthscsa.edu
3    Department of Surgery, Henry Ford Health, Detroit, MI 48202, USA; hjohans1@hfhs.org
*    Correspondence: iokerek1@hfhs.org

**Abstract:** Lung cancer is the most common cancer killer in the world. The standard of care for surgical treatment of non-small cell lung cancer has been lobectomy. Recent studies have identified that sublobar resection has non-inferior survival rates compared to lobectomy, however. Sublobar resection may increase the number of patients who can tolerate surgery and reduce postoperative pulmonary decline. Sublobar resection appears to have equivalent results to surgery in patients with small, peripheral tumors and no lymph node disease. As the utilization of segmentectomy increases, there may be some centers that perform this operation more than other centers. Care must be taken to ensure that all patients have access to this modality. Future investigations should focus on examining the outcomes from segmentectomy as it is applied more widely. When employed on a broad scale, morbidity and survival rates should be monitored. As segmentectomy is performed more frequently, patients may experience improved postoperative quality of life while maintaining the same oncologic benefit.

**Keywords:** lung cancer; sublobar; epidemiology

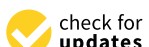



## 1. Introduction

### 1.1. Incidence

Lung cancer is the second most prevalent malignancy in the world, with non-small cell lung cancer (NSCLC) constituting an estimated 85% of all cases [1]. The median age at diagnosis is 71 years [2]. The United States has observed a decrease in the overall incidence of lung cancer, likely related to decreased smoking rates [3]. Although men have traditionally had higher rates of this disease, the gender ratio is now almost equal. Notwithstanding this decline, lung cancer incidence is highest in Black males in the United States. These disparities are likely a result of decreased access to healthcare and social determinants of health [4,5]. Globally, the highest incidence of lung cancer is found in developed nations with lower rates observed in less developed regions. However, these disparities may be partly attributed to under-reporting in regions lacking a robust, centralized reporting system [6]. Cigarette smoking is believed to account for over 80 percent of lung cancers globally [7]. Yet, a notable proportion of lung cancer cases are diagnosed in individuals who have never smoked [8].

### 1.2. Mortality

Despite advances in treatment, lung cancer is the leading cause of cancer-related mortality in the United States, with a death toll surpassing the combined total of the next three most prevalent cancers [9,10]. The five-year survival rates for NSCLC are nearly the lowest among solid organ tumors. Overall, about 18 percent of patients are alive 5 years after their diagnosis [11]. Recent advancements have resulted in improvements in survival rates, particularly evident when comparing two-year versus five-year survival. From 1997

to 2018, the two-year survival for women with NSCLC improved from 36% to 54%, while for men it increased from 28% to 43% [12].

### 1.3. Staging

The early stages of disease present opportunities for curative surgical intervention, potentially leading to improved longevity [13]. Especially in the early stages, lung sparing techniques utilizing sublobar resections are viable options [14]. Traditionally, a lung lobectomy has been regarded as the standard treatment [15]. However, for patients with compromised pulmonary reserves, a lobectomy could result in a significant reduction in pulmonary function [16]. Sublobar resection may also increase the number of patients who can tolerate surgery. The chosen course of treatment considers the patient's overall health, the stage and type of the cancer, and the patient's preferences. Therefore, adherence to meticulous staging processes and judicious patient selection is vital in optimizing outcomes.

### 1.4. Workup

A computed tomography (CT) scan of the chest is commonly the initial step in the diagnostic workup of NSCLC. It aids in detecting the primary tumor, establishing its size, location and relationship to adjacent anatomical structures [17]. Additionally, CT scans facilitate the assessment of lymph nodes and detection of distant metastases, integral to defining the extent of the disease [18]. When mediastinal or hilar lymphadenopathy is observed on CT scan, further invasive mediastinal staging is often recommended [19]. Procedures such as endobronchial ultrasound-guided biopsy or mediastinoscopy aid in ruling out nodal involvement, which would preclude sublobar resection [20]. The decision to perform invasive mediastinal evaluation prior to surgical resection should be based on clinical factors, tumor size and the suspicion of nodal disease on preoperative imaging. Positron emission tomography-computed tomography (PET-CT) is also very important, not only to assess the primary tumor but also to evaluate for locoregional and distant metastases. PET-CT is useful in many different ways. PET-CT can be used as another point of evidence in the decision to operate or not for a patient with a solitary pulmonary nodule. PET-CT results should not be used as the sole criteria in the decision to operate, but instead should be used in combination with other clinical and patient factors. Additionally, PET-CT can help to target specific mediastinal lymph node stations when performing mediastinal lymph node evaluation.

An important part of this evaluation process is assessing the patient's respiratory reserve using pulmonary function tests, including spirometry and diffusion capacity of the lung for carbon monoxide (DLCO) [21]. These tests gauge the operative risk and help determine the appropriate surgical approach. For patients with marginal pulmonary function, sublobar resection may be favored over lobectomy to spare lung parenchyma [22]. If a more detailed evaluation of pulmonary function is needed, testing, such as maximal voluntary ventilation and quantitative ventilation/perfusion scans, may be beneficial. Comprehensive cardiac function assessment and the evaluation of comorbid conditions are also essential components of the preoperative workup. Patients with multiple comorbidities and poor functional status may benefit more from a lung-sparing sublobar resection than a standard lobectomy [23].

Low-dose computed tomography (LDCT) screening is an important topic to discuss in relation to the detection of early stage lung cancer. The National Lung Screening Trial demonstrated a significant reduction in mortality with screening based on age and history of tobacco exposure [24]. LDCT is currently recommended by the United States Preventative Services Task Force for patients between the ages of 50 and 80, with at least a 20 pack-year smoking history and who are currently smoking or have stopped within the last 15 years [25]. The use of LDCT has been shown to detect lung cancers at earlier stages once screening programs are implemented [26]. As LDCT becomes more utilized for eligible individuals, the number of early stage lung cancers that are detected is expected

to rise (Figure 1). As such, there may be increasing opportunities for sublobar operations, such as segmentectomy.

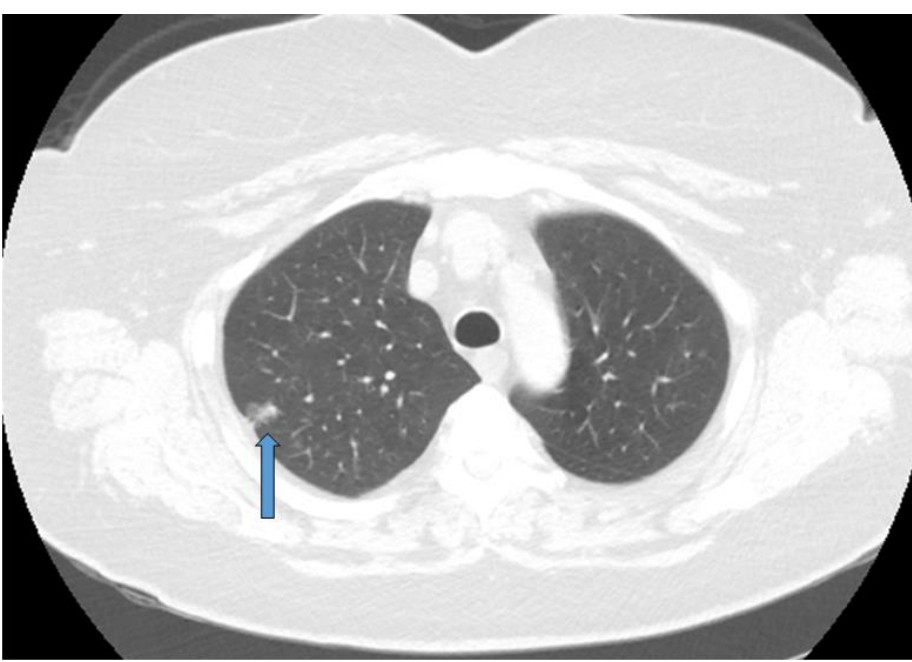

**Figure 1.** Early stage NSCLC detected during screening and treated with segmentectomy. Blue arrow marks the tumor.

## 2. Discussion

### 2.1. Historical Treatment Patterns

This review examined relevant studies from 1995 to present concerning the role of sublobar resection in the treatment of lung cancer. Unlike small cell lung cancer (SCLC), which is typically more sensitive to radiation and chemotherapy, multiple stages of NSCLC have an improved survival when surgery is part of the treatment plan [27]. The proclivity of NSCLC tumors to attain larger dimensions, paired with their tendency for extended localization, underscores their amicability to surgical intervention. This distinct behavior differentiates them from their small-cell counterparts. The size of the lesion plays a critical role in determining an optimal surgical approach. Recent literature analyzing the correlation of survival with tumor size has demonstrated no significant difference in lung cancer-specific survival for NSCLC patients with smaller tumor sizes. When comparing tumor sizes less than 2 cm, a study showed that there was no survival difference between lobectomy and sublobar resection, which includes segmentectomy and wedge resection [28,29]. Furthermore, pulmonary function after 6 months was improved in the sublobar group. Notably, lobectomy has been the standard-of-care surgical approach for stage IA NSCLC since 1995, following the publication of results from a landmark clinical trial [30]. The randomized controlled trial compared lobectomy versus limited resection for T1 N0 NSCLC, and its findings played a significant role in establishing lobectomy as the preferred treatment approach. The anatomy and architecture of the lung allows for the complete removal of the tumor along with a wide resection of lymphatic drainage with lobectomy. This mechanism is likely what leads to decreased local recurrence for tumors in general. However, these recent studies may indicate that such a wide resection is not needed for tumors of less than 2 cm. Regardless of the type of surgery selected, a thorough mediastinal lymph node assessment should be performed and is beneficial for staging and potential therapeutic planning [31]. The adoption of lobectomy as the standard of care is bolstered by robust clinical evidence. While the aforementioned landmark trial by Ginsberg and colleagues found lobectomy to offer superior local control and overall survival compared to sublobar resection for stage I NSCLC, there were several limitations. Its lengthy

duration of seventeen years reflects the challenges in surgical trials. The observed survival differences did not reach statistical significance, possibly due to an underpowered sample size. The older measures used to assess pulmonary function may have missed nuanced differences, and more advanced techniques available today might have offered better insights. Higher recurrence rates with limited resection suggest potential micrometastases beyond the visible tumor, an issue that current advanced imaging and nodal sampling techniques might address. Finally, the relative lack of minimally invasive approaches during the study may limit its relevance, as minimally invasive techniques are associated with less morbidity during surgery. Nevertheless, these recent studies seem to suggest that lobectomy is no longer needed for smaller, node-negative lung cancers.

### 2.2. Newer Evidence Supporting Segmentectomy for Small Cancers

Over the past few years, a growing body of evidence from several studies has illustrated that outcomes for NSCLC patients with tumors of 2 cm or less are similar to lobectomy or sublobar resection (Table 1). A multi-center, international randomized trial was recently published concerning this topic. In this study, patients with T1aN0 NSCLC were randomized to receive either lobectomy or sublobar resection [28]. Their research, spanning a median follow-up period of 7 years, showed the non-inferiority of sublobar resection to lobectomy regarding disease-free survival in patients with tumors of less than 2 cm. The investigators reported equivalent rates of locoregional and distant recurrence between the two surgical groups and noted a slight improvement in pulmonary function six months after surgery in the sublobar resection cohort.

**Table 1.** Recent studies demonstrating similar survival between lobectomy and segmentectomy.

| Title | Authors | Year |
|---|---|---|
| Lobar or Sublobar Resection for Peripheral Stage IA Non–Small-Cell Lung Cancer | Altorki et al. [28] | 2023 |
| Segmentectomy versus lobectomy in small-sized peripheral non-small-cell lung cancer (JCOG0802/WJOG4607L): a multicentre, open-label, phase 3, randomised, controlled, non-inferiority trial | Saji et al. [32] | 2022 |
| Survival outcomes in a prospective randomized multicenter Phase III trial comparing patients undergoing anatomical segmentectomy versus standard lobectomy for non-small cell lung cancer up to 2 cm | Stamatis et al. [33] | 2022 |
| Equivalent Survival Between Lobectomy and Segmentectomy for Clinical Stage IA Lung Cancer | Onaitis et al. [34] | 2020 |
| Sublobar resection is comparable to lobectomy for screen-detected lung cancer | Kamel et al. [22] | 2021 |
| Recurrence and Survival Outcomes After Anatomic Segmentectomy Versus Lobectomy for Clinical Stage I Non–Small-Cell Lung Cancer: A Propensity-Matched Analysis | Landreneau et al. [35] | 2014 |
| Comparison of Lobectomy and Sublobar Resection for Stage IA Elderly NSCLC Patients (≥70 Years): A Population-Based Propensity Score Matching's Study | Zhang et al. [29] | 2021 |
| Sublobar resection versus lobectomy for stage I non-small cell lung cancer: an appropriate choice in elderly patients? | Fiorelli et al. [36] | 2016 |
| Sublobar resection is equivalent to lobectomy for T1a non-small cell lung cancer in the elderly: a Surveillance, Epidemiology, and End Results database analysis | Razi et al. [37] | 2016 |
| Lobar and sub-lobar lung resection in octogenarians with early stage non-small cell lung cancer: factors affecting surgical outcomes and long-term results | Dell'Amore et al. [38] | 2015 |
| Early lung cancer in the elderly: sublobar resection provides equivalent long-term survival in comparison with lobectomy | Liu et al. [39] | 2014 |
| Sublobar resection is associated with better perioperative outcomes in elderly patients with clinical stage I non-small cell lung cancer: a multicenter retrospective cohort study | Zhang et al. [40] | 2019 |
| Local control and short-term outcomes after video-assisted thoracoscopic surgery segmentectomy versus lobectomy for pT1c pN0 non-small-cell lung cancer | Forster et al. [41] | 2023 |
| Sublobar resection provides an equivalent survival after lobectomy in elderly patients with early lung cancer | Okami et al. [42] | 2010 |
| Perioperative mortality and morbidity after sublobar versus lobar resection for early-stage non-small-cell lung cancer: post-hoc analysis of an international, randomised, phase 3 trial (CALGB/Alliance 140503) | Altorki et al. [43] | 2018 |

A separate multicenter, randomized trial compared the outcomes of lobectomy and segmentectomy in over 1000 patients with clinical stage IA NSCLC [32]. The study revealed a survival advantage favoring segmentectomy over lobectomy. Interestingly, there was no clinically meaningful difference in postoperative respiratory function in the segmentectomy group compared to the lobectomy group, which was contrary to initial expectations. These two rigorous, randomized controlled trials underscore the potential benefits of sublobar

resection for patients with small, peripheral NSCLC without clinical evidence of nodal disease.

Moreover, non-randomized studies lend further support to the comparable survival outcomes between lobectomy and segmentectomy for patients with clinical stage IA NSCLC. In a propensity-matched analysis of the Society of Thoracic Surgeons General Thoracic Surgery Database linked with Medicare data, investigators found similar survival patterns in patients over 65 years undergoing either lobectomy or sublobar resection for clinical stage IA NSCLC [34]. Surprisingly, this parity persisted even in patients who were pathologically N0, despite a higher rate of upstaging observed in the lobectomy group. In a different investigation, investigators examined the National Lung Screening Trial database and found no significant difference in survival between lobectomy and sublobar resection among patients with screen-detected lung cancer [22]. Additionally, several studies have examined the role of sublobar resection in elderly patients or those with limited pulmonary reserve, all of which reported survival rates akin to those following lobectomy [29,36–38,40,44,45]. Considering these findings together with robust randomized controlled trial data, the application of sublobar resection emerges as a viable strategy for early stage NSCLC. This strategy is particularly pertinent as the use of low-dose CT screening continues to rise, increasing the likelihood of detecting early stage lung cancer.

Patients who have tumors of less than or equal to 2 cm and no nodal disease should be considered for segmentectomy if technically feasible. Wedge resection is generally a less technically challenging surgery than segmentectomy and may have a role in some pulmonary tumors. In particular, wedge resection offers similar survival to segmentectomy for pulmonary metastases [46]. However, wedge resection appears to have inferior overall survival when compared with segmentectomy for early stage, node-negative primary lung cancers [47]. When possible, segmentectomy should be considered for primary early stage NSCLC versus wedge resection.

### 2.3. Challenges of Segmentectomy

Anatomic segmentectomy has been proposed as an alternative to lobectomy in stage I NSCLC patients, specifically those at an increased risk of lobectomy-associated complications [48]. However, the inherent complexity and significant variability of the segmentectomy procedure pose challenges. These challenges may potentially result in incomplete resections, inadequate margins and suboptimal lymph node sampling [35]. Additionally, anatomical factors such as the irregular shape and location of certain lung segments can make complete resection particularly challenging. Pulmonary segmentectomy can be a technically involved operation. The anatomical pattern of some segments is more complex than other areas, increasing the difficulty of the operation. In addition, the identification of the intersegmental plane is important to achieve adequate margins and reduce the rates of postoperative air leak.

The use of segmentectomy will likely increase the number of patients with early lung cancer who are candidates for surgery, especially those patients with poor lung function [49]. While some research has indicated the better short-term preservation of lung function with segmentectomy [50], the landmark Lung Cancer Study Group trial found no significant difference in pulmonary function at 1.5 years after operation.

Assessing lymph nodes during segmentectomy can also be challenging due to often inadequate hilar and mediastinal lymph node dissection [51]. This might result in the understaging of patients with node-positive disease who could potentially benefit more from lobectomy and adjuvant chemotherapy. There is the potential utility of segmentectomy for small, peripheral stage I NSCLC tumors in selecting high-risk patients for whom lobectomy is not feasible. However, lobectomy should be the approach selected for larger tumors or the presence of lymphadenopathy [52].

There are several new technologies that may facilitate segmentectomy now and in the future. Augmented reality (AR) and three-dimensional printing have been used to guide surgery for segmentectomy and subsegmentectomy [53]. The printed three-dimensional

models are used by surgeons preoperatively to plan their surgical conduct and determine the planes between segments. In particular, three-dimensional printing can help surgeons to see the precise locations and branch points of the pulmonary vascular and bronchial structures. In the operating room, AR can provide significant magnification and an augmented overlay of the vascular and bronchial anatomy to aid the dissection during surgery. Although the use of AR and three-dimensional printing is relatively infrequent now, we expect that these modalities will be increasingly used as segmentectomy becomes more common.

Fluorescence imaging can also help to determine segmental planes during surgery. Use of indocyanine green (ICG) fluorescence can demarcate the appropriate area for resection during segmentectomy [54]. ICG is injected into the pulmonary vein, followed by near-infrared fluorescence imaging during thoracoscopy. This imaging can help surgeons to transect the lung parenchyma in the correct area and allow for the resection of the entire segment.

### 2.4. Disparities in Lung Cancer Treatment

Existing literature unequivocally suggests that minority populations are, stage for stage, less frequently offered surgical treatments for lung cancer [55]. Consequently, it is critical to contemplate whether an increased rate of segmentectomy procedures would alleviate or exacerbate this inequity. Many of these underrepresented populations experience obstacles to care related to social determinants of health, which have been associated with delayed presentations of disease [56]. The social determinants of health can present obstacles to care for several reasons. Financial constraints, lack of transportation, insurance status and lack of understanding can lead to less use of standard of care in disadvantaged communities. As segmentectomy is used more frequently, disparities in access to this surgical approach may arise.

Socioeconomic status (SES) is a strong predictor of patient outcomes. SES is independently associated with mortality rates in stage I NSCLC patients, regardless of surgical intervention, race or marital status [57]. A previous study reported that high-income patients and White patients had an increased likelihood of undergoing surgical treatment and experienced a higher five-year survival rate [58]. Such findings were echoed in a comprehensive meta-analysis of 94 studies, which revealed a positive correlation between income and lung cancer survival [59].

Health insurance status and racial factors have a marked effect on the likelihood of a patient receiving surgery for early stage NSCLC. Patients without private insurance and African American patients were significantly less likely to undergo lobectomy [60]. Additionally, in a study assessing patients with NSCLC enrolled in clinical trials, higher education predicted longer overall survival [61].

Previous literature has shown that according to the United States Preventive Services Task Force guidelines, eligibility for lung cancer screening was less likely among Black and Hispanic patients, thereby perpetuating disparities in lung cancer care [62]. These findings are consistent with another study that found that the highest five-year survival rate for NSCLC is in White patients and individuals from higher SES levels [63]. Other literature has reported worse overall survival rates in early stage lung cancer among non-Hispanic Black individuals compared to non-Hispanic White individuals, emphasizing the necessity to improve surgical access for minority populations [64].

Evidence from the National Lung Cancer Screening Trial (NLST) highlighted the importance of appropriate follow-up for early intervention, with African American patients demonstrating significantly lower follow-up rates compared to White patients [65]. Notably, Black patients, those with Medicaid and those with housing insecurity were all more likely to miss their low-dose screening appointments, highlighting the need for targeted interventions by governmental and healthcare policymakers to address these inequities in adherence [66]. Furthermore, Black males with clinical stage I NSCLC were nearly 30 percent less likely to undergo surgery compared to their White counterparts [67].

In a review of the National Cancer Database, a study found that patients with stage I NSCLC experiencing social determinants of health (SDOH) disparities had lower utilization of surgery, higher open surgery proportions, higher 90-day mortality and lower median survival [68]. Within the Veterans Health Administration system, Black patients undergoing surgical resection for clinical stage I NSCLC were less likely to receive adequate lymph node sampling but exhibited significantly better risk-adjusted overall survival compared to White patients [69]. As segmentectomies become more common, it will be important to determine whether disparities in lung cancer treatment patterns improve or worsen.

### 2.5. Potential Negative Consequences of Widespread Segmentectomy Adoption

The distribution of surgical procedures for NSCLC, including segmentectomies, across different healthcare centers is largely influenced by factors such as institutional resources, surgeon experience and the center's patient population characteristics [70,71]. High-volume centers and academic institutions tend to perform more segmentectomies compared to their counterparts, although the reasons for this are multifactorial and complex [72]. High-volume centers have more opportunities for surgical procedures, leading to increased experience and potentially improved patient outcomes [73]. This higher volume could facilitate better familiarity with complex procedures such as segmentectomies, thereby increasing the likelihood of their use. Moreover, these centers are often equipped with superior facilities and are more likely to be at the forefront of surgical advancements, enhancing their ability to perform such technical procedures [74].

Academic institutions, due to their research-driven nature and the responsibility to train new surgeons, also tend to perform a higher number of complex surgical procedures such as segmentectomies. These institutions often pioneer new techniques and are typically more inclined to adopt innovative surgical approaches sooner [75]. Furthermore, the presence of surgical residents and fellows who are learning these new techniques may promote the use of segmentectomy. However, this does not imply that segmentectomy is universally suitable or preferred in all high-volume or academic centers. The decision to perform a segmentectomy versus a lobectomy is contingent upon various factors, such as the patient's overall health, tumor location, tumor size and expertise of the surgeon [14,76]. Even within these centers, there may be variations in segmentectomy utilization based on the individual surgeon's comfort with the procedure [77]. It is critical to note that a higher number of performed procedures does not inherently equate to better patient outcomes. Quality of care, thoroughness of patient evaluation and proper postoperative follow-up are essential to ensure optimal surgical outcomes [78]. Additionally, centers with less experience in segmentectomy should be encouraged to collaborate and learn from their high-volume and academic counterparts to improve their skills and patient care. Collaboration among institutions will likely help to create a standardization for operation and broadly improve the level of the uniformity of segmentectomy. In addition, collaboration may help to decrease complication rate in centers that are adopting the use of segmentectomy for the treatment of early stage NSCLC.

### 2.6. Outcomes to Assess in the Future

As we explore the increased use of segmentectomies in the treatment of NSCLC, it is critical to monitor key outcomes for the benefit of patient care and equity. First, we should track postoperative morbidity and mortality, emphasizing the reduction in complications and death rates within the first 30 days following surgery and thereafter. This can serve as an indicator of the safety of the procedure. One significant complication of note is postoperative air leak. Segmentectomy has been shown to be associated with a higher rate of air leak after surgery [79]. Postoperative air leak is not inconsequential, as it is associated with increased length of stay, increased index hospitalization costs and increased costs after discharge [80].

Second, it is crucial to assess the oncologic effectiveness of segmentectomies. Metrics, such as recurrence rate, progression-free survival and overall survival, should be closely

monitored. Previous literature has shown that the number of lymph nodes harvested during segmentectomy is less than during lobectomy [81]. Although survival is statistically similar, data should be analyzed as the prevalence of segmentectomy increases. Additionally, the incorrect location of the segmental planes may lead to incomplete resection and increased local recurrence rates. These outcomes will have to be studied in upcoming years to evaluate the effectiveness of segmentectomy as it becomes the treatment of choice for early non-small cell lung cancer. For many years, lobectomy was the standard of care, as it was shown to be associated with better long-term outcomes [82]. However, more recent data listed above suggest that currently segmentectomy has equivalent survival to lobectomy.

We should also observe the impact of segmentectomies on lung function preservation, comparing postoperative pulmonary function tests with preoperative baselines and contrasting outcomes following lobectomy [83].

Equity of access and outcomes must be examined as well, ensuring that benefits are not influenced by race, socioeconomic status or institutional capabilities. Regular assessments of segmentectomy distribution and outcomes across various demographics will help us identify and rectify disparities [84]. Lastly, we need to evaluate institutional patterns of segmentectomy utilization, encouraging cross-institutional collaboration for skill development and improved patient care [85].

### *2.7. Limitations*

There are some limitations of this review. While a thorough literature review was performed, there are numerous further studies regarding this topic and the older literature supported lobectomy as having improved outcomes. Additionally, the majority of studies referenced in this review are retrospective and subject to selection bias. Nevertheless, it appears that segmentectomy is a suitable operation for early stage NSCLC with equal oncologic results.

### 3. Conclusions

The evolving role of segmentectomy for peripheral, stage I NSCLC presents potential benefits in terms of decreased morbidity, improved pulmonary function and the option for resection in patients who would not otherwise be surgical candidates. Our discussion has focused on segmentectomy as the preferred sublobar resection of choice. However, challenges such as surgical complexity and adequate lymph node assessment remain. Racial and socioeconomic disparities persist in lung cancer surgical treatments, with minority and low-income populations receiving less surgical intervention and subsequently facing worse outcomes. Institutional factors, including surgical volume and teaching status, also influence segmentectomy usage.

### 4. Future Directions

The continuous monitoring of specific outcomes is critical to refining our surgical approach for stage I NSCLC, including patient survival, access to care and institutional surgical patterns. An observed increase in recurrence rates may prompt a re-evaluation of this strategy. The ultimate goal is to enhance patient outcomes and ensure equitable treatment access, requiring ongoing research into the intersection of patient factors, surgical expertise, institutional resources and social determinants of health.

**Author Contributions:** P.B. and J.T. contributed to the literature review and data collection. P.B., J.T., H.H. and I.C.O. contributed to the writing of the manuscript. P.B., J.T. and I.C.O. contributed to the formal analysis of the manuscript. I.C.O. provided supervision and project administration for the manuscript. All authors have read and agreed to the published version of the manuscript.

**Funding:** This research received no external funding.

**Conflicts of Interest:** The authors declare no conflict of interest.

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
