# Peer review of "The Role of Sublobar Resection for the Surgical Treatment of Non-Small Cell Lung Cancer"

_curroncol, doi:10.3390/curroncol30070509_

Round 1

Reviewer 1 Report

Thank you very much for inviting me to review the article titled “The role of sublobar resection for the surgical treatment of non-small cell lung cancer”.

In their article, the authors make a detailed review of the literature on the very important and currently hot issue of segmentectomy.

The article is written in very good quality English. The structure of the article is correct, it contains all necessary sections.

I only have a few comments about the article:

1. Positron emission tomography–computed tomography (PET-CT) is very important for lung cancer staging. When discussing the issue of preoperative workup (subsection 1.4), this examination should also be mentioned.

2. Low-dose CT (LDCT) screening has been found to reduce lung cancer mortality. Because LDCT increases the detection of the early-stage lung cancer, it may be particularly important for increasing the percentage of segmentectomies. I suggest adding a sentence or two to subsection 1.4 on this topic.

3. There are many new surgical techniques that can facilitate segmentectomy. If deemed appropriate, the authors may consider discussing the use of indocyanine green fluorescence, robotic surgery, 3D reconstruction and augmented reality in relation to segmentectomy.

4. Segmentectomy is a difficult operation, as the authors discuss in their manuscript. Incorrectly performed segmentectomy may result in postoperative complications, incorrect nodal staging, incomplete resection and worse long-term results. I believe these issues should be discussed more extensively and made more explicit in the manuscript.

Once again, I would like to congratulate you on an interesting and detailed review on segmentectomy.

Author Response

Reviewer 1

Comment

Positron emission tomography–computed tomography (PET-CT) is very important for lung cancer staging. When discussing the issue of preoperative workup (subsection 1.4), this examination should also be mentioned.

Response

We agree that the role of PET-CT should be included.  In section 1.4 we have added “Positron emission tomography-computed tomography (PET-CT) is also very important, not only to assess the primary tumor, but also to evaluate for locoregional and distant metastases. PET-CT is useful in many different ways. PET-CT can be used as another point of evidence in the decision to operate or not for a patient with a solitary pulmonary nodule. PET-CT results should not be used as the sole criteria in the decision to operate, but instead should be used in combination with other clinical and patient factors. Additionally, PET-CT can help to target specific mediastinal lymph node stations when performing mediastinal lymph node evaluation. “

Comment

Low-dose CT (LDCT) screening has been found to reduce lung cancer mortality. Because LDCT increases the detection of the early-stage lung cancer, it may be particularly important for increasing the percentage of segmentectomies. I suggest adding a sentence or two to subsection 1.4 on this topic.

Response

This is an important point.  We have added to section 1.4 “Low-dose computed tomography (LDCT) screening is an important topic to discuss in relation to the detection of early stage lung cancer. The National Lung Screening Trial demonstrated a significant reduction in mortality with screening based on age and history of tobacco exposure [24]. LDCT is currently recommended by the United States Preventative Services Task Force for patients between the ages of 50 and 80, with at least 20 pack-year smoking history and are currently smoking or have stopped within the last 15 years [25]. Use of LDCT has been shown to detect lung cancers at earlier stages once screening programs are implemented [26]. As LDCT becomes more utilized for eligible individuals, the number of early-stage lung cancers that are detected is expected to rise. As such, there may be increasing opportunities for sublobar operations.” We have also added multiple references.

Comment

There are many new surgical techniques that can facilitate segmentectomy. If deemed appropriate, the authors may consider discussing the use of indocyanine green fluorescence, robotic surgery, 3D reconstruction and augmented reality in relation to segmentectomy.

Response

These modalities are being used more frequently and we agree that it is good to discuss some of these newer techniques. In section 3 we have added “There are several new technologies that may facilitate segmentectomy now and in the future. Augmented reality (AR) and 3-dimensional printing have been used to guide surgery for segmentectomy and subsegmentectomy [48]. The printed 3-dimensional models are used by surgeons preoperatively to plan their surgical conduct and determine the planes between segments. In particular, 3-dimensional printing can help surgeons to see the precise locations and branch points of the pulmonary vascular and bronchial structures. In the operating room, AR can provide significant magnification and an augmented overlay of the vascular and bronchial anatomy to aid the dissection during surgery. Alhough the use of AR and 3-dimensional printing is relatively infrequent now, we expect that these modalities will be increasingly used as segmentectomy becomes more common.” We have also added “Fluorescence imaging can also help to determine segmental planes during surgery. Use of indocyanine green (ICG) fluorescence can demarcate the appropriate area for resection during segmentectomy [49]. ICG is injected into the pulmonary vein, followed by near-infrared fluorescence imaging during thoracoscopy. This imaging can help surgeons to transect the lung parenchyma in the correct area and allow for resection of the entire segment.”

Comment

Segmentectomy is a difficult operation, as the authors discuss in their manuscript. Incorrectly performed segmentectomy may result in postoperative complications, incorrect nodal staging, incomplete resection and worse long-term results. I believe these issues should be discussed more extensively and made more explicit in the manuscript.

Response

This is an important point and we have expanded our discussion on multiple ideas.  Regarding postoperative complications, we have added “One significant complication of note is postoperative air leak. Segmentectomy has been shown to be associated with a higher rate of air leak after surgery [75]. Postoperative air leak is not inconsequential, as it is associated with increased length of stay, increased index hospitalization costs and increased costs after discharge [76].” We have also added “Previous literature has shown that the number of lymph nodes harvested during segmentectomy is less than during lobectomy [77]. Although survival is statistically similar, data should be analyzed as the prevalence of segmentectomy increases. Also, incorrect location of the segmental planes may lead to incomplete resection and increased local recurrence rates. These outcomes will have to be studied in upcoming years to evaluate the effectiveness of segmentectomy as it becomes the treatment of choice for early non-small cell lung cancer.”

Reviewer 2 Report

Thank you for this insightful review.

I have some comments/questions:

1) In workup take out high resolution CT that is more for ILD, just say CT chest with IV contrast and most will know that they are thin 2.5 mm cuts. 

2) can we use a different term other than part 5, maybe section? 

3) most is on segmentectomy? I think the title should be changed or made clear that you are mostly discussing the role of segmentectomy in early stage NSCLC 

also would say for early stage NSCLC 

Author Response

Reviewer 2

Comment

In workup take out high resolution CT that is more for ILD, just say CT chest with IV contrast and most will know that they are thin 2.5 mm cuts. 

Response

We agree and have removed “high resolution.”

Comment

Can we use a different term other than part 5, maybe section?

Response

We have changed every “Part” to “Section.”

Comment

Most is on segmentectomy? I think the title should be changed or made clear that you are mostly discussing the role of segmentectomy in early stage NSCLC 

Response

We tend to agree with you on this point. In the conclusion we have added the following to clarify our position “Our discussion has focused on segmentectomy as the preferred sublobar resection of choice.”

Comment

Also would say for early stage NSCLC 

Response

We agree and we have added the following to section 1.4 “the number of early-stage lung cancers that are detected is expected to rise. As such, there may be increasing opportunities for sublobar operations such as segmentectomy.

Reviewer 3 Report

The authors submitted a review concerning the sublobar resections for the treatment of lung cancer. The paper deals with a very interesting topic, still debated in the scientific field. Some revisions are necessary to improve the quality of the paper and to clarify some concepts.

I have some suggestions:

- the time range of search of the articles taken into account in this review should be reported 

- Wedge resection and segmentectomy present different outcomes and should be proposed in selected patients. This argument should be discussed in the review, reporting the different results related to sublobar anatomic and non-anatomic resection, reported in the literature. Furthermore, the selection criteria for sublobar resection should be discussed, according to guidelines and literature

- Discussion is divided into 6 parts called "part 1", "part 2"...This subdivision has not a clear reason: the renomination of each section about its argument could be clarifying. For example, part 4 could be titled "Disparities in lung cancer treatment"

- Lobectomy results associated with better long-term outcomes in some studies. Please, report this information, debating this conclusion

- The limit of the review should be discussed

Author Response

Dear editorial board and staff,

Thank you for reviewing our manuscript entitled "The Role of Sublobar Resection for the Surgical Treatment of Non-Small Cell Lung Cancer.” The reviewers made excellent suggestions and requested minor revisions.  We have responded to each comment below.  Thank you.

Reviewer 1

Comment

 The time range of search of the articles taken into account in this review should be reported

Response

In the first paragraph (1.1) we have added “This review examined relevant studies from 1995 to present concerning the role of sublobar resection in the treatment of lung cancer.”

Comment

Wedge resection and segmentectomy present different outcomes and should be proposed in selected patients. This argument should be discussed in the review, reporting the different results related to sublobar anatomic and non-anatomic resection, reported in the literature. Furthermore, the selection criteria for sublobar resection should be discussed, according to guidelines and literature

Response

We agree that adding this section will give readers a clear understanding about the guidelines in which to consider segmentectomy.  We have added “Patients who have tumors less than or equal to 2 centimeters and no nodal disease should be considered for segmentectomy if technically feasible. Wedge resection is generally a less technically challenging surgery than segmentectomy and may have a role in some pulmonary tumors. In particular, wedge resection offers similar survival to segmentectomy for pulmonary metastases [42]. But wedge resection appears to have inferior overall survival when compared with segmentectomy for early-stage, node-negative primary lung cancers [43]. When possible, segmentectomy should be considered for primary early-stage NSCLC versus wedge resection.”

Comment

Discussion is divided into 6 parts called "part 1", "part 2"...This subdivision has not a clear reason: the renomination of each section about its argument could be clarifying. For example, part 4 could be titled "Disparities in lung cancer treatment"

Response

We have named each section as suggested with descriptive titles as opposed to numbering each section.

Comment

Lobectomy results associated with better long-term outcomes in some studies. Please, report this information, debating this conclusion

Response

We agree that it is important to mention that less current literature did support lobectomy as having better outcomes.  In the second paragraph of the “Outcomes to assess in the future” section, we have added “For many years lobectomy was the standard of care, as it was shown to be associated with better long-term outcomes [80]. But more recent data listed above suggest that currently segmentectomy has equivalent survival to lobectomy.”

Comment

The limit of the review should be discussed

Response

We agree that a Limitations section is important for the overall perspective.  We have added a Limitations section immediately prior to the Conclusions.  We have added “There are some limitations of this review. While there was a thorough literature review performed, there are numerous studies regarding this topic and older literature supported lobectomy as having improved outcomes. Additionally, the majority of studies referenced in this review are retrospective and subject to selection bias. Nevertheless, it appears that segmentectomy is a suitable operation for early-stage NSCLC with equal oncologic results.”

Round 2

Reviewer 3 Report

Congratulation, the paper is well-written and its focus is now clear.

I have only a suggestion:

The sentence "This review examined relevant studies from 1995 to present concerning the role of sublobar resection in the treatment of lung cancer" should be moved to the first part of the Discussion, in which the Authors introduce the aim of the review and the methods of its execution. 

Author Response

Thank you for the response.  We have moved the sentence to the first part of the discussion as requested.  

我们还根据要求回复了该电子邮件,并为所有作者提供了机构电子邮件地址。